# LEARNING TO OBSERVE WITH REINFORCEMENT LEARNING

## ABSTRACT

We consider a decision making problem where an autonomous agent decides on which actions to take based on the observations it collects from the environment. We are interested in revealing the information structure of the observation space illustrating which type of observations are the most important (such as position versus velocity) and the dependence of this on the state of agent (such as at the bottom versus top of a hill). We approach this problem by associating a cost with collecting observations which increases with the accuracy. We adopt a reinforcement learning (RL) framework where the RL agent learns to adjust the accuracy of the observations alongside learning to perform the original task. We consider both the scenario where the accuracy can be adjusted continuously and also the scenario where the agent has to choose between given preset levels, such as taking a sample perfectly or not taking a sample at all. In contrast to the existing work that mostly focuses on sample efficiency during training, our focus is on the behaviour during the actual task. Our results illustrate that the RL agent can learn to use the observation space efficiently and obtain satisfactory performance in the original task while collecting effectively smaller amount of data. By uncovering the relative usefulness of different types of observations and trade-offs within, these results also provide insights for further design of active data acquisition schemes.

## 1 INTRODUCTION

Autonomous decision making relies on collecting data, i.e. observations, from the environment where the actions are decided based on the observations. We are interested in revealing the information structure of the observation space illustrating which type of observations are the most important (such as position versus velocity). Revealing this structure is challenging since the usefulness of the information that an observation can bring is a priori unknown and depends on the environment as well as the current knowledge state of the decision-maker, for instance, whether the agent is at the bottom versus the top of a hill and how sure the agent is about its position. Hence, we're interested in questions such as "Instead of collecting all available observations, is it possible to skip some observations and obtain satisfactory performance?", "Which observation components (such as the position or the velocity) are the most useful when the object is far away from (or close to) the target state?". The primary aim of this work is to reveal this information structure of the observation space within a systematic framework.

We approach this problem by associating a cost with collecting observations which increases with the accuracy. The agent can choose the accuracy level of its observations. Since cost increases with the accuracy, we expect that the agent will choose to collect only the observations which are most likely to be informative and worth the cost. We adopt a reinforcement learning (RL) framework where the RL agent learns to adjust the accuracy of the observations alongside learning to perform the original task. We consider both the scenario where the accuracy can be adjusted continuously and also the scenario where the agent has to choose between given preset levels, such as taking a sample perfectly or not taking a sample at all. In contrast to the existing work that mostly focuses on sample efficiency during training, our focus is on the behaviour during the actual task. Our results illustrate that the RL agent can learn to use the observation space efficiently and obtain satisfactory performance in the original task while collecting effectively smaller amount of data.

## 2 RELATED WORK

A related setting is active learning (Settles, 2010; Donmez et al., 2010) where an agent decides which queries to perform, i.e., which samples to take, during training. For instance, in an active learning set-up, an agent learning to classify images can decide which images from a large dataset it would like to have labels for in order to have improved classification performance. In a standard active learning approach (Settles, 2010; Donmez et al., 2010) as well as its extensions in RL (Lopes et al., 2009), the main aim is to reduce the size of the training set, hence the agent tries to determine informative queries during training so that the performance during the test phase is optimal. In the test phase, the agent cannot ask any questions; instead, it will answer questions, for instance, it will be given images to label. In contrast, in our setting the agent continues to perform queries during the test phase, since it still needs to collect observations during the test phase, for instance as in the case of collecting camera images for an autonomous driving application. From this perspective, one of our main aims is to reduce the number of queries the agent performs during this actual operation as opposed to number of queries in its training phase.

Another related line of work consists of the RL approaches that facilitate efficient exploration of state space, such as curiosity-driven RL and intrinsic motivation (Pathak et al., 2017; Bellemare et al., 2016; Mohamed & Rezende, 2015; Still & Precup, 2012) or active-inference based methods utilizing free-energy (Ueltzhöffer, 2018; Schwöbel et al., 2018); and the works that focus on operation with limited data using a model (Chua et al., 2018; Deisenroth & Rasmussen, 2011; Henaff et al., 2018; Gal et al., 2016). In these works, the focus is either finding informative samples (Pathak et al., 2017) or using a limited number of samples/trials as much as possible by making use of a forward dynamics model (Boedecker et al., 2014; Chua et al., 2018; Deisenroth & Rasmussen, 2011; Henaff et al., 2018; Gal et al., 2016) during the agent's training. In contrast to these approaches, we would like to decrease the effective size of the data or the number of samples taken during the test phase, i.e. operation of the agent after the training phase is over.

Representation learning for control and RL constitutes another line of related work (Watter et al., 2015; Hafner et al., 2019; Banijamali et al., 2018). In these works, the transformation of the observation space to a low-dimensional space is investigated so that action selection can be performed using this low-dimensional space. Similar to these works, our framework can be also interpreted as a transformation of the original observation space where an effectively low-dimensional space is sought after. Instead of allowing a general class of transformations on the observations, here we consider a constrained setting so that only specific operations are allowed, for instance, we allow dropping some of the samples but we do not allow collecting observations and then applying arbitrary transformations on them.

Our work associates a cost with obtaining observations. Cost of data acquisition in the context of Markov decision processes (MDPs) has been considered in a number of works, both as a direct cost on the observations (Hansen, 1997; Zubek & Dietterich, 2000; 2002) or as an indirect cost of information sharing in multiple agent settings (Melo & Veloso, 2009; De Hauwere et al., 2010). Another related line of work is performed under the umbrella of configurable MDPs (Metelli et al., 2018; Silva et al., 2019) where the agent can modify the dynamics of the environment. Although in our setting, it is the accuracy of the observations rather than the dynamics of the environment that the agent can modify, in some settings our work can be also interpreted as a configurable MDP. We further discuss this point in Section 4.2.

## 3 PROPOSED FRAMEWORK AND THE SOLUTION APPROACH

### 3.1 PRELIMINARIES

Consider a Markov decision process given by $\langle \mathcal{S}, \mathcal{A}, \mathcal{P}, R, P_{s_0}, \gamma \rangle$ where $\mathcal{S}$ is the state space, $\mathcal{A}$ is the set of actions, $\mathcal{P} : \mathcal{S} \times \mathcal{A} \times \mathcal{S} \to \mathbb{R}$ denotes the transition probabilities, $R : \mathcal{S} \times \mathcal{A} \to \mathbb{R}$ denotes the bounded reward function, $P_{s_0} : \mathcal{S} \to \mathbb{R}$ denotes the probability distribution over the initial state and $\gamma \in (0, 1]$ is the discount factor.

The agent, i.e. the decision maker, observes the state of the system $s_t$ at time $t$ and decides on its action $a_t$ based on its policy $\pi(s, a)$. The policy mapping of the agent $\pi(s, a) : \mathcal{S} \times \mathcal{A} \to [0, 1]$ is possibly stochastic and gives the probability of taking the action $a$ at the state $s$. After the agent

implements the action $a_t$, it receives a reward $r(s_t, a_t)$ and the environment moves to the next state $s_{t+1}$ which is governed by $\mathcal{P}$ and depends on $a_t$ and $s_t$. The aim of the RL agent is to learn an optimal policy mapping $\pi(s, a)$ so that the expected return, i.e. expected cumulative discounted reward, $J(\pi) = \mathbb{E}_{a_t \sim \pi, s_t \sim P}[\sum_t \gamma^t r(s_t, a_t)]$ is maximized.

## 3.2 PARTIAL OBSERVABILITY

Although most RL algorithms are typically expressed in terms of MDPs, in typical real-life applications the states are not directly observable, i.e., the observations only provide partial, possibly inaccurate information. For instance, consider a vehicle which uses the noisy images with limited angle-of-view obtained from cameras mounted on the vehicle for autonomous-driving decisions. In such scenarios, the data used by the agent to make decisions is not a direct representation of the state of the world. Hence, we consider a partially observable Markov decision process (POMDP) where the above MDP is augmented by $\mathcal{O}$ and $\mathcal{P}_o$ where $\mathcal{O}$ represents the set of observations and $\mathcal{P}_o : \mathcal{S} \rightarrow \mathcal{O}$ represents the observation probabilities. Accordingly, the policy mapping is now expressed as $\pi(o, a) : \mathcal{O} \times \mathcal{A} \rightarrow [0, 1]$.

The observation vector at time $t$ is given by $o_t = [o_t^1; \ldots; o_t^n] \in \mathbb{R}^n$, where $n$ is the dimension of the observation vector. The observations are governed by

$$o_t \sim p_o(o_t | s_t; \beta_t) \tag{1}$$

where $p_o(o_t | s_t; \beta_t)$ denotes the conditional probability distribution function (pdf) of $o_t$ given $s_t$ and is parametrized by the accuracy vector

$$\beta_t = [\beta_t^1; \ldots; \beta_t^n] \in \mathbb{R}^n \tag{2}$$

The parameter $\beta_t^i \geq 0$ represents the average accuracy of the observation component $i$ at time step $t$, i.e. $o_t^i$. For instance, say we have two observations, position $o^1$ and velocity $o^2$. Then, $\beta_t^1$ denotes the accuracy of the position and $\beta_t^2$ denotes the accuracy of the velocity. As $\beta_t^i$ increases, the accuracy of the observation $o_t^i$ decreases. Given $s_t$ and $\beta_t$, the observations are statistically independent, i.e. we have the factorization

$$p_o(o_t | s_t; \beta_t) = \prod_{i=1,\ldots,n} p_{o^i}(o_t^i | s_t; \beta_t^i) \tag{3}$$

where $p_{o^i}(o_t^i | s_t; \beta_t^i)$ denotes the conditional pdf of $o_t^i$ given $s_t$ and $\beta_t^i$.

Note that $\beta_t^i$ determines the average accuracy, i.e. the accuracy in the statistical sense. We provide an example below:

**Example:** Consider the common Gaussian additive noise model with

$$o_t^i = s_t^i + v_t^i, \qquad i = 1, \ldots, n, \tag{4}$$

where $s_t = [s_t^1; \ldots; s_t^n] \in \mathbb{R}^n$ is the state vector and $v_t = [v_t^1; \ldots; v_t^n] \in \mathbb{R}^n$ is the Gaussian noise vector with $\mathcal{N}(0, \text{diag}(\sigma_{v_t^i}^2))$. Here, $v_t$ and $v_{t'}$ are statistically independent (stat. ind.) for all $t \neq t'$ and also $v_t$ and $s_{t'}$ are stat. ind. for all $t, t'$. Under this observation model, a reasonable choice for $\beta_t^i$ is $\beta_t^i = \sigma_{v_t^i}^2$. Hence, we parametrize $p_o^i(.)$ as $p_o^i(o_t^i | s_t^i; \beta_t^i) = \mathcal{N}(s_t^i, \beta_t^i = \sigma_{v_t^i}^2)$. Note that the parametrization in terms of $\beta_t^i$ can be done in multiple ways, for instance, one may also adopt $\beta_t^i = \sigma_{v_t^i}$.

## 3.3 DECISION MAKER CHOOSES THE ACCURACY OF THE OBSERVATIONS

The agent can choose $\beta_t^i$, hence $\beta_t^i$ is a decision variable. Observations have a cost which increases with increasing accuracy, i.e. the cost increases with decreasing $\beta_t^i$.

- In Scenario A, the agent can vary $\beta_t^i$ on a continuous scale, i.e. $\beta_t^i \in [0, \infty]$.
- In Scenario B, the agent chooses between i) collecting all the observations with a fixed level of accuracy or ii) not getting any of them at all. This setting corresponds to the case with $\beta_t = \bar{\beta}_t \mathbf{1}$, $\bar{\beta}_t \in \{\beta_F, \infty\}$, where $\mathbf{1} \in \mathbb{R}^n$ denotes the vector of ones. Here $\beta_F \geq 0$ represents a fixed accuracy level. Note that $\beta_F$ can be zero, corresponding to the case $o_t = s_t$.

**Remark 3.1** Our proposed setting can be interpreted as a constrained representation learning problem for RL. In particular, consider the problem of learning the best mapping $h(.)$ with

$$z_t = h(\bar{o}_t) \tag{5}$$

from the high-dimensional original observations $\bar{o}_t$ to some new possibly low-dimensional variables $z_t$ so that control can be performed reliably on $z_t$ instead of $\bar{o}_t$. Such settings have been utilized in various influential work, see for instance E2C approach of Watter et al. (2015).

The proposed approach can be also formulated in a representation framework. In particular, we interpret the possibly noisy observations $o_t$ as the effectively low-dimensional representation $z_t$ used in (5). Hence, consider the mapping $\bar{h}(.)$

$$o_t = \bar{h}(\bar{o}_t), \tag{6}$$

where $o_t$ and $\bar{o}_t$ denote the noisy and the original measurements, respectively. Compared to (5), the family of the mappings allowed in (6) is constrained, i.e. one can only adjust the accuracy parameter instead of using arbitrary transformations from $\bar{o}_t$ to $o_t$. Here, $o_t$ is effectively low-dimensional compared to $\bar{o}_t$ because i) noise decreases the dynamic range and allows effectively higher compression rates for the data (Scenario A); or ii) the total number of observations acquired is smaller (Scenario B). Note that not all transformations from $s_t$ to $o_t$ can be written using (6) as an intermediate step. From this perspective, the formulation in (1) can be said to be more general than (6).

### 3.3.1 MOTIVATION

The primary motivation behind the proposed framework is to reveal the inherent nature of the observation space in terms of usefulness of information the observations provide with respect to the task at hand. The secondary motivation is to provide a RL framework for solving decision making problems when the observations have a cost associated with them.

In regard to the first task, we note the following: To reveal this information structure, we associate an artificial cost with the observations that increase with the accuracy. Hence, only the observation components (or the observation vectors) which are mostly likely to be informative and worth the cost will be collected. This decision heavily depends on the state that the agent believes itself to be in. For instance, in the case of balancing an object at an unstable state (such as pendulum in OpenAi Gym (Brockman et al., 2016)), we intuitively expect that the agent does not need accurate measurements when it is far away from the target state. Hence, we're interested in questions such as "Is it possible to skip some observations and obtain satisfactory performance?", "Which observation components (such as the position or the velocity) are most useful when the object is far away from (or close to) the target state?", "How are these results affected by the possible discrepancy between the true state the agent is in and the one that it believes it to be in due to noisy or skipped observations?". The proposed framework reveals this information structure within a systematic setting.

In regard to the second task, we note that there are many practical problems where there is a cost associated with acquiring observations (Hansen, 1997; Zubek & Dietterich, 2000; 2002), for instance consider the expensive medical tests (i.e. observations) that have to performed to diagnose a certain disease (Zubek & Dietterich, 2002) and wireless communications where there is a cost associated with channel usage (i.e. the right to use a communication channel) and a power cost that increases with the reliability of communications (Goldsmith, 2005; Cover & Thomas, 1991), see also Section A.1. The proposed framework can be used to find efficient observation strategies in such problems and to quantify the possible performance degradation due to the observation cost.

**Examples:** The proposed scenarios A and B also correspond to practical data acquisition schemes. We now give some examples: An example for Scenario A is the case where the observations are obtained using different sensors on the device where the accuracy of each sensor can be individually adjusted. Another example is the case where the sensors are distributed over the environment and the readings of the sensors has to be relayed to central decision unit using individual compression of each observation type and wireless communications. Here, the compression and the wireless communication introduces an accuracy-cost trade-off where the agent can choose to operate at different points of. Please see Section A.1 for an example illustrating the accuracy-cost trade-off in

wireless communications. An example for Scenario B is the remote control of a device, such as a drone, where all sensor readings of the device are compressed together and then sent to a decision unit. Since all readings are compressed and transmitted together, a decision of whether to transmit the whole observation vector or not has to be made, for instance due the limited power or wireless channel occupancy constraints.

### 3.4 REWARD SHAPING

Reward shaping is a popular approach to direct RL agents towards a desired goal. Here, we want the agent not only move towards the original goal (which is encouraged by the original reward $r$), we also want it to learn to control $\beta_t$. Hence, we propose reward shaping in the following form:

$$\tilde{r}_t = f(r_t, \beta_t) \tag{7}$$

where $r_t$ is the original reward, $\tilde{r}_t$ is the new modified reward and $f(r_t, \beta_t)$ is a monotonically non-decreasing function of $r_t$ and $\beta_t^i$, $\forall i$. Hence, the agent not only tries to maximize the average of the original reward but it also tries to maximize the "inaccuracy" of the measurements. This can be equivalently interpreted as minimizing the cost due to accurate measurements. In the case where there is a direct cost function $c^i(.)$ that increases with the accuracy of the observation $o^i$ (see, for instance, the example in Section A.1 where transmission power can be interpreted as the direct cost), the following additive form can be used

$$\tilde{r}_t = r_t - \lambda \sum_{i=1}^{n} c^i(\beta_t^i), \tag{8}$$

where $c^i(\beta_t^i)$ is a non-increasing function of $\beta_t^i$ and $\lambda \geq 0$ is a weighting parameter. Hence, the agent's aim is to maximize the original reward as well as minimize the cost of the observations.

## 4 EXPERIMENTS

### 4.1 SETTING

**Observation Models:** We consider the following environments from the OpenAI Gym (Brockman et al., 2016): MountainCarContinuous-v0, Pendulum-v0, CartPole-v1. In this section, we illustrate how the modified environment with noisy observations is obtained for MountainCarContinuous-v0. The details and the parameter values for the other environments can be found in the Appendix A.2. We also consider a version of MountainCarContinuous-v0 with observations of the vertical position, which is presented in Section A.4.

We first explain Scenario A, and then Scenario B. The original observations of the mountain car environment are the position $x_t$ and the velocity $\dot{x}_t$. In our framework, the agent has access to noisy versions of these original observations

$$\tilde{x}_t = x_t + Q_x \times \Delta x_t(\beta_t^1), \tag{9a}$$

$$\tilde{\dot{x}}_t = \dot{x}_t + Q_{\dot{x}} \times \Delta \dot{x}_t(\beta_t^2), \tag{9b}$$

where $\Delta x_t(\beta_t^1) \sim \mathcal{U}(-\beta_t^1, \beta_t^1)$, $\Delta \dot{x}_t(\beta_t^2) \sim \mathcal{U}(-\beta_t^2, \beta_t^2)$ and $\mathcal{U}(-\beta, \beta)$ denotes the uniform distribution over $[-\beta, \beta]$. The noise variables are stat. ind., in particular $\Delta x_t(\beta_t^1)$ and $\Delta \dot{x}_t(\beta_t^2)$ are stat. ind. from each other and also stat. ind. over time. Here, $Q_x$ and $Q_{\dot{x}}$ determine the ranges of the noise levels and they are set as the $0.1$ times of the full range of the corresponding observation, i.e., $Q_x = 0.18$ and $Q_{\dot{x}} = 0.014$.

Our agent chooses $\beta_t^i \in [0, 1]$ in addition to the original action of the environment, i.e. the force $a_t$ that would be exerted on the car. The original reward of the environment per step is given by $r_t = -0.1 \times a_t^2$. The reward is shaped using an additive model

$$\tilde{r}_t = r_t + \kappa_A \times \left( \frac{1}{n} \sum_{i=1}^{n} \beta_t^i \right), \tag{10}$$

where $n = 2$ and $\kappa_A > 0$ is chosen as $5 \times 10^{-6}$. The original environment has also a termination reward which the agent gets when the car passes the target position at $0.45$, which is also provided to our agent upon successful termination.

Table 1: Comparison of the average returns

| ENVIRONMENT | ORIGINAL | A | B |
|---|---|---|---|
| MOUNTAINCARCONTINUOUS-V0 | 94 | 94 | 94 |
| PENDULUM-V0 | -152 | -158 | -170 |
| CARTPOLE-V1 | 494 | 482 | 472 |

In Scenario B, at each time instant we either have no observation or we obtain the original observation vector, i.e. $\tilde{x}_t = x_t$ and $\tilde{\dot{x}}_t = \dot{x}_t$. These cases correspond to $\bar{\beta}_t = \infty$ and $\bar{\beta}_t = 0$, respectively. The reward function is given as $\tilde{r}_t = r_t + \kappa_B \times g(\bar{\beta}_t)$ where $\kappa_B = 0.5$; and $g(\bar{\beta}_t) = -1$ for $\bar{\beta}_t = 0$, and $0$ otherwise. In the implementation, we have mapped $\infty$ to 1, i.e. the decision variable is $\bar{\beta}_t \in \{0, 1\}$, hence $\bar{\beta}_t = 1$ corresponds to not obtaining a sample in Scenario B.

**RL algorithm:** We adopt a deep RL setting, combining reinforcement learning with deep learning using the policy-based approach Trust Region Policy Optimization (TRPO) (Schulman et al., 2015; Hill et al., 2018). The parameters are kept constant for all experiments and are provided in Appendix A.3. For Scenario A, at each time step, noisy observations obtained at that time step are fed to the algorithm as the observations. For Scenario B, the last acquired observation is fed to the algorithm as the observation at that time step.

**Plots:** Unless otherwise stated, all results are reported as averages (such as average cumulative rewards and average $\beta_t^i$) using 1000 episodes. For the plots, observation space is mapped to a grid with uniform intervals. Averages are taken with respect to the number of visits to each given range of the observation state. For example, for Scenario A the average of $\beta_t^i$ when $\tilde{x}_t \in [-0.1, +0.1]$ is shown as one average value at the center $0$. For Scenario B, we report the sample skip frequency, i.e. the number of times the agent decided not to acquire a new observation when the last observed state of the agent falls into a given interval, such as the average sample skip frequency for $\tilde{x} \in [-0.1, +0.1]$ is reported as one value at $0$. In all 2-D plots, the color pink indicates there was no visit to that observation state.

## 4.2 OVERVIEW

We benchmark our results against the performance of the agent that use the original observations, and trained using the same RL algorithm. The resulting average cumulative rewards in terms of $r_t$ are presented in Table 1. We present the reward corresponding only to the original task so that we can evaluate the success of the agent in this task. These results illustrate that the agent can learn to adjust the accuracy level and still obtain successful performance. For the Mountain car environment, all agents have the same average return and for the others, the agents working with the noisy/skipped observations have a slightly weaker performance but still achieve the task of bringing/keeping the pendulum/pole in a vertical position in a reasonable number of time steps.

At first sight, it may be surprising that the agent can learn to perform these tasks satisfactorily even if we have not injected any memory to our algorithm, for instance when we only use the current noisy observations for Scenario A. On the other hand, note that in these environments the observations are either noisy versions of hidden states which govern the dynamics or they are closely related to them. From the point of the agent that treats the noisy observations as state this can be interpreted as a configurable MDP (Metelli et al., 2018; Silva et al., 2019) where the agent controls the noise of the dynamics. Hence, the task of the agent can be interpreted as adjusting the noise level in the dynamics which does not necessarily require usage of memory in the decision maker.

We now focus on the data collection strategies chosen by the agent for the mountain car and pendulum environments. The results for the other environments are provided in the appendix.

## 4.3 MOUNTAIN CAR

The chosen noise levels and the sample skip frequencies for the mountain car environment are presented in Figure 1-2. Note that in Figure 1c, we present the sample skip frequency with respect to the velocity and the position on the same plot, where the legend also gives the corresponding $x$-axis label. In the mountain car environment, the car starts randomly around position $-0.5$ and it has to

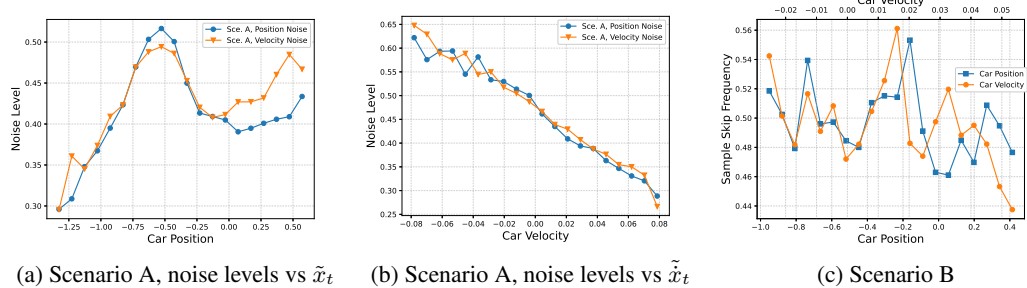

(a) Scenario A, noise levels vs $\tilde{x}_t$     (b) Scenario A, noise levels vs $\tilde{\tilde{x}}_t$     (c) Scenario B

Figure 1: Mountain car, noise levels or sample skip frequency versus one observation type

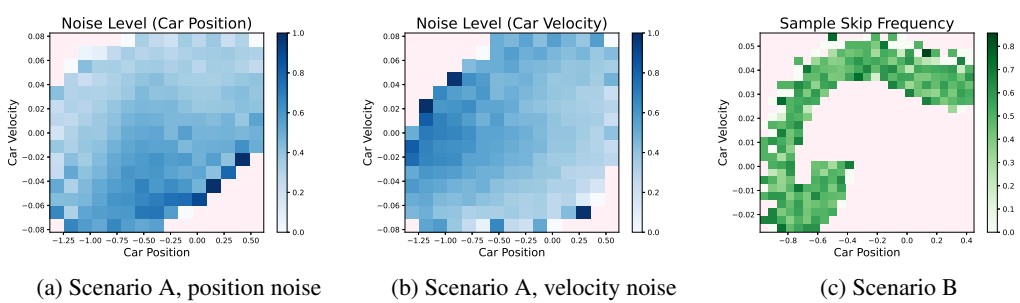

(a) Scenario A, position noise     (b) Scenario A, velocity noise     (c) Scenario B

Figure 2: Mountain car, noise levels or skip frequencies over the whole observation space

first go in the reverse direction (corresponding to a negative velocity) to climb the hill located around position $-1.25$ in order to gain momentum and climb to hill at the right (corresponding to a positive velocity) and reach the target location $0.45$ which is at the top of this hill. The results reflect some of the trade-offs in this strategy:

Figure 1a shows that most noisy observations in position and velocity (Scenario A) are preferred around $-0.5$ (where the car position is initialized), and the most accurate samples are taken when the car is around position $-1.2$. This is the position where the car has to make sure that it has reached to the top of the left hill so that it has enough momentum to climb the right hill. In the case of the dependence of the noise level on the velocity, Figure 1b shows that accurate samples are preferred when the velocity has high positive values. We note that this is not the only viable observation strategy and there are multiple observation strategies that give approximately the same average return in the original task. These can be explored using different $Q$ and $\kappa$ values in our framework.

Figure 1c shows that approximately half of the samples are dropped in Scenario B regardless of the observation state, suggesting a high inherent sampling rate in the environment. This difference in the behaviour with the noisy and skipped observations illustrates the fundamental difference in these frameworks. In the case of noisy observations, the agent has to discover that the observations are uncertain and counteract this uncertainty. On the other hand, when taking perfect observations are possible, as in the case of Scenario B, the agent can internalize the exact environment dynamics (since mountain car environment has no inherent noise in its observations) and determine its exact state using the previous observed state and its action.

Comparing Figure 2a-2b with Figure 2c, we observe that in the case of noisy observations a larger part of observation space is visited, which is partly due the fact that the plots are drawn according to the observations acquired by the agent and not the true states. Note that this does not affect the performance in the original task, as illustrated in Table 1.

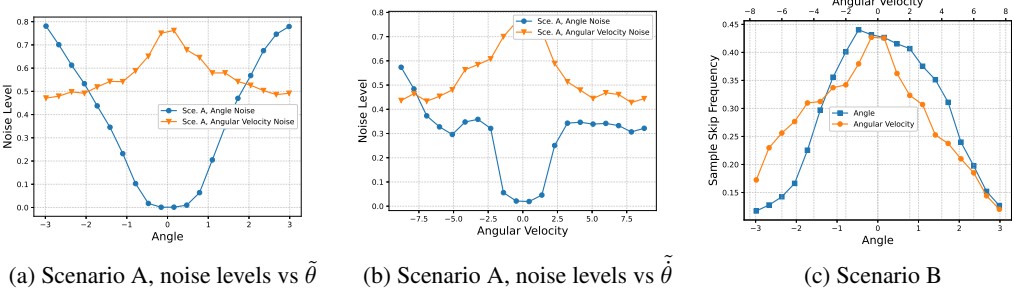

(a) Scenario A, noise levels vs $\tilde{\theta}$  (b) Scenario A, noise levels vs $\dot{\tilde{\theta}}$  (c) Scenario B

Figure 3: Pendulum, noise levels or sample skip frequency versus one observation type

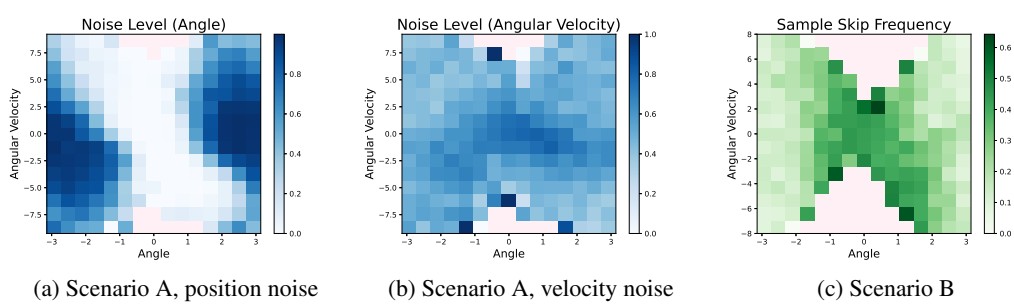

(a) Scenario A, position noise  (b) Scenario A, velocity noise  (c) Scenario B

Figure 4: Pendulum, noise levels or skip frequencies over the whole observation space

## 4.4 PENDULUM

The results for the pendulum are presented in Figure 3-4. Here, the task is to keep the pendulum at a vertical position, corresponding to an angle of 0. Figure 3a and Figure 4a show that observations with low position (i.e. angle) noise (Scenario A) are preferred when the pendulum is close to the vertical position and has relatively small angular velocity. On the other hand, when the samples can be completely skipped (Scenario B), the agent skips a large ratio of the samples in this region, as shown in Figure 3c and Figure 4c. Note that the agent spends most of the episode in this target region in the vertical position. Here, the agent prefers noiseless samples since a noisy sample may cause the control policy to choose a wild movement which might destabilize the pendulum. On the other hand, the agent may safely skip some of the samples at the upright position as the last sample is very close to current one because the angular velocity is typically low.

## 5 DISCUSSION AND CONCLUSIONS

We have proposed a framework for revealing the information structure of the observation space in a systematic manner. We have adopted a reinforcement learning approach which utilizes a cost function which increases with the accuracy of the observations. Our results uncover the relative usefulness of different types of observations and the trade-offs within; and provide insights for further design of active data acquisition schemes for autonomous decision making. Further discussion of our results and some research directions are as follows:

- Our results illustrate that settings with the inaccurate observations and skipped observations should be treated differently since the type of uncertainty that the agent has to counteract in these settings are inherently different.

- Strategies for processing of the noisy/skipped observations should be investigated. Questions such as the following arise: " Should all the processing be off-loaded to the RL agent or

should the pre-processing of observations be performed, similar to Kalman filtering in the case of linear control under linear state space models (Ljung, 1999)?", "How does the answer to the former question depend on the RL approach, the environment and the observation models?"

- Our results suggest that inherent sampling rate of some of the standard RL environments may be higher than needed (for instance, see the Mountain Car environment where on average one can skip one out of every two samples without affecting the performance), indicating yet-another reason why some of these environments are seen as unchallenging for most of the state-of-art RL algorithms.

- We have provided a quantification of the sensitivity of the agent's performance to noisy/skipped observations at different observation regions illustrating that this sensitivity can be quite different based on the observation region. Utilizing this information for supporting robust designs as well as preparing adversarial examples is an interesting line of future research.

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

## A APPENDIX

### A.1 EXAMPLE: WIRELESS COMMUNICATIONS

We now provide a motivating example to illustrate how observations can have a cost that is increasing with the accuracy and the decision maker can choose this accuracy level.

A standard model for single terminal wireless communications is the additive white Gaussian noise (AWGN) channel (Goldsmith, 2005; Cover & Thomas, 1991)

$$y_t = x_t + v_t \tag{11}$$

where $x_t$ represents the channel input (i.e. message at the transmitter ) at time $t$, $y_t$ represents the corresponding channel output (i.e. the observation at the receiver) and the white Gaussian random process $v_t$ represents the channel noise. The capacity of this channel, i.e. the maximum number of information bits that can be sent, is determined by the signal-to-noise ratio (SNR), i.e. the average power in $x_t$ divided by the average power in $v_t$. In particular, the capacity is given by (Goldsmith, 2005; Cover & Thomas, 1991)

$$C = \log_2(1 + \frac{P_x}{P_v}) \tag{12}$$

where $P_x$ and $P_v$ are the average power levels of $x_t$ and $v_t$, respectively. Hence, the capacity increases with $P_x$. On the other hand, one cannot use a very high value of $P_x$ since broadcasting at high power levels is costly. In particular, $P_x$ directly contributes to the actual power required by the transmitter. Note that $P_x$ controls the accuracy of the observations. In particular, by dividing both sides by $\sqrt{P_x}$, (11) can be equivalently represented as

$$\bar{y}_t = \bar{x}_t + \bar{v}_t \tag{13}$$

where $\bar{y}_t \triangleq \frac{1}{\sqrt{P_x}} y_t$, $\bar{x}_t \triangleq \frac{1}{\sqrt{P_x}} x_t$ and $\bar{v}_t \triangleq \frac{1}{\sqrt{P_x}} v_t$. The average power of $\bar{x}_t$ is 1 and average power of $\bar{v}_t$ is $P_v/P_x$. The SNR, and hence, the channel capacity are the same in (11) and (13) and hence these representations are equivalent for all relevant purposes. In particular, determining $P_x$ directly determines the effective noise level. With $v_t$ Gaussian, we have $v_t \sim \mathcal{N}(0, P_v)$. Hence, the conditional distribution of the observations $\bar{y}_t$ is given by $p(\bar{y}_t | \bar{x}_t) = \mathcal{N}(\bar{x}_t, P_v/P_x)$ where $P_v/P_x$ can be chosen as $\beta_t$. Hence, as the accuracy of the observations increases ($P_v/P_x$ decreases ), the cost of the observations ($P_x$) increases. In this context, several interesting questions that relates to the accuracy of the observations and the power cost can be posed, for instance how to distribute a certain total power budget $P_{total}$ over channels $y_t^i = x_t^i + v_t^i$ with different intrinsic power levels $P_{v_i}$.

This example illustrates the basic premise of our problem setting in a practical scenario; a decision maker who can adjust the noise levels of the observations which has a cost associated with them. It also suggests that the constraints on the wireless communications constitute a general and potential hindrance in remote control applications. Consider a device that makes the observations and takes actions but gets its commands (i.e. decisions about which actions to take) from another decision unit, such as the control of a robot or a drone by a remotely run RL algorithm which is controlling a large number of such units. Here, it is beneficial to consider policies that can work with inaccurate observations since sending accurate measurements are costly from a power perspective, which will be particularly important for a device with a limited battery, such as a drone flying at a remote location. Similarly, if the wireless communication channel cannot be used at all times, for instance, due to the limited bandwidth available, RL methods that can utilize the limited communication resources efficiently and optimize performance under such conditions are needed.

### A.2 ENVIRONMENT PARAMETERS

In this section, we provide the parameters for all the environments in the experiments that are used directly from OpenAI Gym. We also consider a vertical position version of MountainCarContinuous-v0, which is explained in Section A.4.

Consider a generic environment with the observation variables $o_t^i$, where $o_t^i$ denotes the $i^{th}$ observation variable at time $t$. The limited-accuracy observations $\tilde{o}_t^i$ are obtained using

$$\tilde{o}_t^i = o_t^i + Q^i \times \Delta o_t^i(\beta_t^i) \tag{14}$$

Table 2: Environment parameters, reward weighting for different scenarios

| ENVIRONMENT | $\kappa_A, \kappa_B$ |
|---|---|
| MOUNTAINCARCONTINUOUS-V0 | $5 \times 10^{-6}, 0.5$ |
| PENDULUM-V0 | $1, 0.2$ |
| CARTPOLE-V1 | $0.2, 0.04$ |

Table 3: Hyperparameters of the TRPO algorithm

| PARAMETER | VALUE |
|---|---|
| GRADIENT DAMPENING FACTOR | 2.35E-05 |
| WEIGHT FOR THE ENTROPY LOSS | 0.01118 |
| GAMMA | 0.98 |
| GAE FACTOR | 0.9 |
| KULLBACK-LEIBLER LOSS THRESHOLD | 0.000193 |
| NO. OF TIMESTEPS TO RUN PER BATCH | 1024 |
| NO. ITERS. FOR LEARNING FOR VALUE FUNC | 10 |
| STEPSIZE OF VALUE FUNC. | 0.00428 |

where $\Delta o_t^i \sim \mathcal{U}(-\beta_t, \beta_t)$. We choose $Q^1 = 0.1$ and $Q^2 = 0.2$ for the Pendulum-v0, $Q^i = 0.2$ for the CartPole-v1, and $Q^i = 0.1$ for the MountainCarContinuous-v0. The ordering of the observations is the same with the ones provided in OpenAI Gym (Brockman et al., 2016). For instance, for MountainCarContinuous-v0, position and velocity correspond to $o^1$ and $o^2$, respectively. Note that indices start with $i = 0$ in OpenAI Gym whereas here we start with $i = 1$.

The reward function under Scenario A is given by

$$\tilde{r}_t = r_t + \kappa_A \times \left( \frac{1}{n} \sum_{i=1}^{n} \beta_t^i \right), \tag{15}$$

where $r_t$ is the original reward and $\kappa_A > 0$. For Scenario B, it is given by $\tilde{r}_t = r_t + \kappa_B \times g(\bar{\beta}_t)$ where $g(\bar{\beta}_t) = -1$ for $\bar{\beta}_t = 0$, and 0 otherwise. The associated $\kappa$ values for different environments are presented in Table 2.

The scaling factor $Q$'s for the noise levels and $\kappa$ values for the reward function are determined empirically by first fixing $Q$ (as a percentage of the full range of the associated observation) and searching for $\kappa$ values that provide satisfactory performance in the original task. Note that the rest of the values are determined by the specifications of the environments in OpenAI Gym. The results depend on the values of $Q$ and $\kappa$. For instance, using larger $\kappa$ puts a larger weight on the reward due to noise. Hence, the agent prioritizes the reward due to noise instead of the reward from the original environment and, for large enough $\kappa$ values, the agent cannot learn to perform the original task.

## A.3 TRPO PARAMETERS

The same TRPO parameters are used in all experiments. These are provided in Table 3.

## A.4 MOUNTAIN CAR WITH OBSERVATIONS OF THE VERTICAL POSITION

To have a better understanding of the effect of partial observability, we have investigated the following modification on MountainCarContinuous-v0: Instead of the horizontal position, the agent uses the vertical position as the observation. Hence, the observations are given by

$$\tilde{y}_t = y_t + Q_y \times \Delta y_t(\beta_t^1), \tag{16a}$$

$$\tilde{\dot{x}}_t = \dot{x}_t + Q_{\dot{x}} \times \Delta \dot{x}_t(\beta_t^2), \tag{16b}$$

where the vertical position $y_t \in [0.1, \ 1]$ is given by $y_t = 0.45 \sin(3x_t) + 0.55$ (Brockman et al., 2016) and $\Delta y_t(\beta_t^1) \sim \mathcal{U}(-\beta_t^1, \beta_t^1)$ and $\Delta \dot{x}_t(\beta_t^2) \sim \mathcal{U}(-\beta_t^2, \beta_t^2)$. Note that due to $\sin(\cdot)$ function,

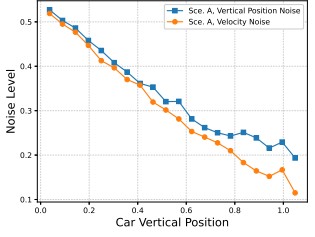

(a) Scenario A, noise levels vs $\tilde{y}_t$

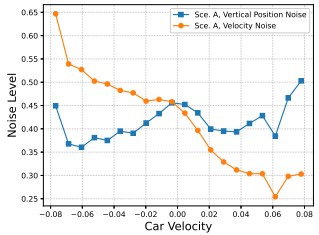

(b) Scenario A, noise levels vs $\tilde{x}_t$

Figure 5: Mountain car with vertical position observation, noise levels versus one observation type

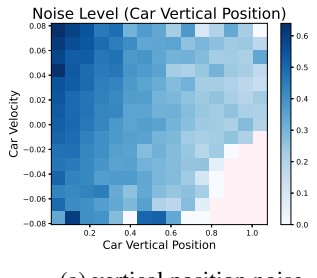

(a) vertical position noise

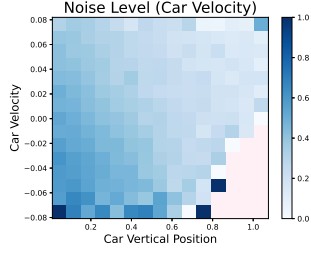

(b) velocity noise

Figure 6: Mountain car with vertical position observation, noise levels over the observation space

for most of the $y_t$ values in the range $[0.1, \ 1]$, there are two possible horizontal position ($x_t$) values. Hence, this environment constitutes a POMDP even without any observation noise. Similar to our experiments with the original environment, $Q_y$ and $Q_{\dot{x}}$ are set as the $0.1$ times of the full range of the corresponding observation, i.e., $Q_x = 0.09$ and $Q_{\dot{x}} = 0.014$. As before, the reward is calculated with (10) with $\kappa_A = 5 \times 10^{-6}$.

The average return due to the original task is 93, hence the agent again learns to perform the original task successfully, see Table 1 for comparison. The chosen noise levels are presented in Figure 5-6. Comparing these results with Figure 1-2 where the agent takes the horizontal position observation, we observe that the general trend of the velocity noise with respect to the velocity are the same in both settings, i.e. decreasing as the agent moves from the negative velocities to positive velocities. Comparing Figure 5 with Figure 1, we observe that lower relative noise levels are preferred for the setting with the vertical location observations.

## A.5 ADDITIONAL RESULTS -CART POLE

We now provide the results for the cart pole environment in Figure 7-10, which were not included in the main text due to page limitations. For the sake of brevity, the noise levels over observations pairs is only provided for the position noise levels whereas averages are provided for all observation types.

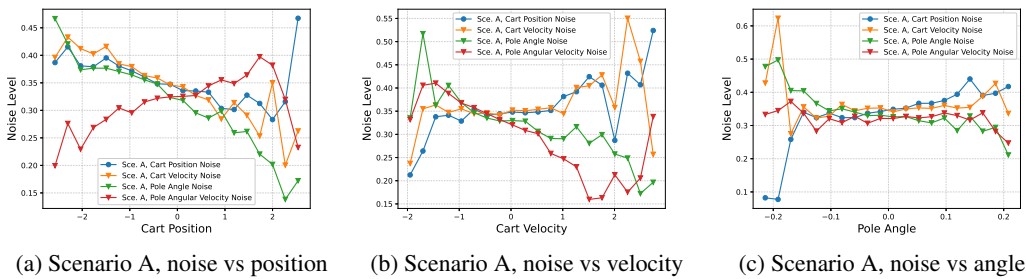

(a) Scenario A, noise vs position  (b) Scenario A, noise vs velocity  (c) Scenario A, noise vs angle

Figure 7: Cart pole, noise levels versus one observation type

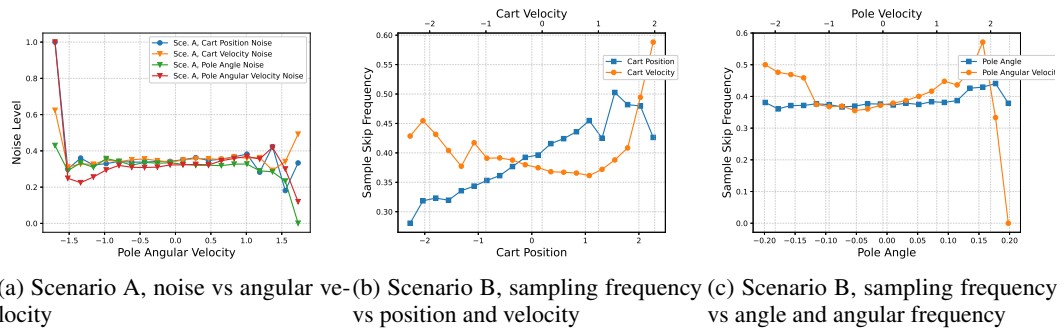

(a) Scenario A, noise vs angular ve- (b) Scenario B, sampling frequency (c) Scenario B, sampling frequency
locity                              vs position and velocity           vs angle and angular frequency

Figure 8: Cart pole, noise levels or sample skip frequency versus one observation type

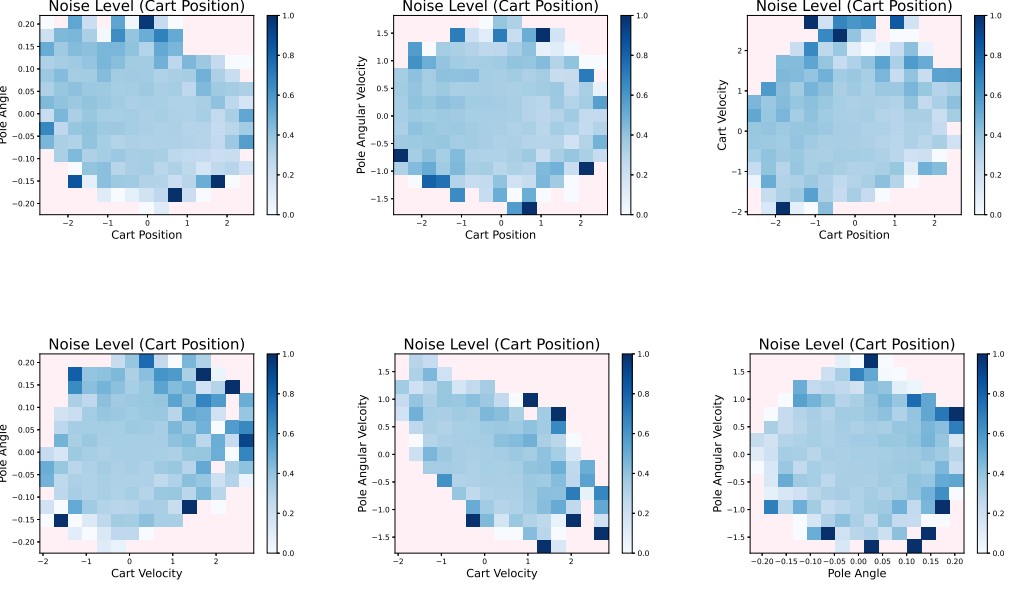

Figure 9: Cart Pole, Scenario A, noise levels for the position over the pairs of the observation variables

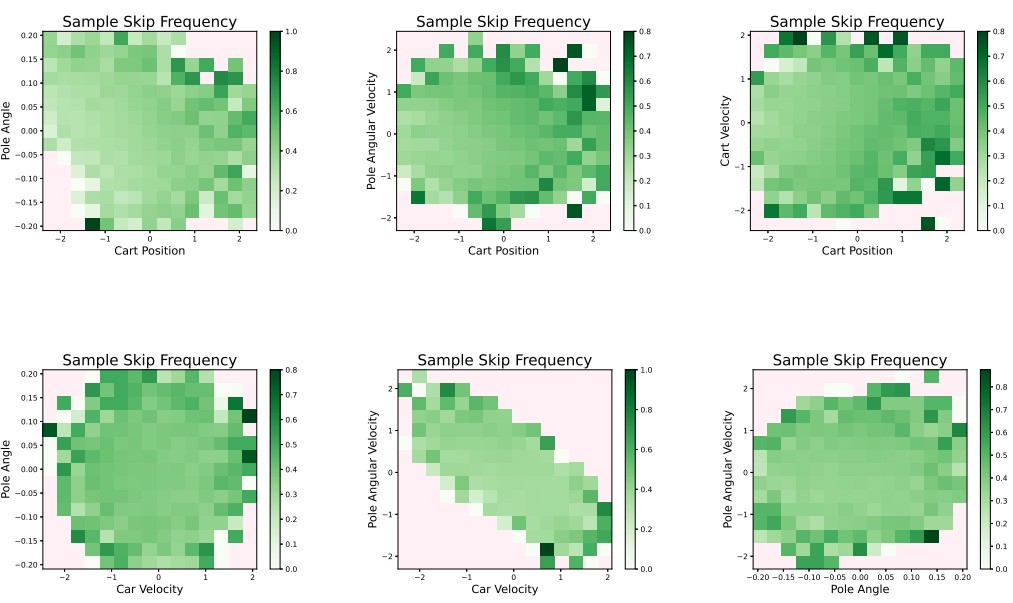

Figure 10: Cart Pole, Scenario B, sample skip frequencies over the pairs of the observation variables

