# OpenReview forum: "Learning to Observe with Reinforcement Learning"
_ICLR.cc/2021/Conference — Reject_

### Official Review · AnonReviewer4 · 2020-10-26
**Small contribution and absent comparisons with previous works**

**Rating:** 4
**Confidence:** 5

**Review:**

This paper aims at studying an optimized way of collecting samples from an environment, discarding the ones for which the accuracy of the observation is high. This way the agent focuses on collecting only the samples that improve the knowledge of the state space.

This paper could be presented better, as the motivations of the work and the description of the method lack clarity and effectiveness. First, the title is somewhat misleading, as we cannot say that the agent is "learning to observe", which can remind something more related to feature extraction in representation learning. Indeed, the agent is learning to explore states under a certain criterion, i.e. minimizing the accuracy of the observation, closely reminding all the literature about intrinsically motivated exploration, that in this paper is only cited in the related works. After all, it looks to me that this paper is exclusively proposing a form of intrinsic reward, but it fails to explain it thoroughly. In particular, only a small subsection, namely 3.4, is dedicated to this description, moreover referring to "reward shaping", which is not the same concept as intrinsic motivation. The experimental section is weak, as it only analyses two simple RL problems, more problematically not comparing with any method in the literature.

Pros
------
* The paper addresses an interesting problem that can potentially improve sample-efficiency in deep RL problems.

Cons
-------
* Poor description of the methodology, in particular explaining its connection with intrinsic motivation;
* No deep RL problems considered;
* No comparisons with methods in literature.

I recommend the authors to substantially restructure the paper to include a better analysis of how their method compares with intrinsic motivation, include deep RL problems where the problem of exploration and accuracy of observations is more accentuated, and add comparisons with representative baselines, e.g. Pathak et al (2017), Bellemare et al (2016), etc..

Post-rebuttal feedback
-------------------------------
I thank the authors for their reply.

> In contrast, our paper focuses on the following question: “how can we reduce the number/accuracy of the samples the agent
takes during the test phase”? (Here, the test phase corresponds to the agent’s behaviour after the training is completed.)

I agree with the authors that intrinsic motivation is different, and perhaps in my review I expressed this concern too strongly. So I thank the authors for their long and informative answer.

> We believe that the reviewer refers to the problems where a possibly large multi-dimensional data such as images in games are used as input to the RL algorithm.

Exactly. Experiments on high-dimensional problems would make the contribution of this paper stronger, considering the rather limited  theoretical/methodological impact that it has now. I strongly suggest the authors to work in this direction, perhaps on robotic application if possible.

After the rebuttal, I still argue for rejection, although I increase my score from 3 to 4.

---

> ### Author Response · Authors · 2020-11-15
> **Response to Reviewer 4**
>
> Thank you for your comments. We believe that  there is some confusion about the position of this paper with respect to the literature on intrinsic motivation. In particular, we do not agree that these works provide direct benchmarks for this paper. We discuss these points below:
>
> This paper and intrinsic motivation:
>
> We agree that this paper is related to the line of work on intrinsic motivation, as illustrated by our inclusion of discussion of these works in the original submission. On the other hand, we believe that these two settings address different questions:
>
> The literature on intrinsic motivation  focuses on sample efficiency during training and addresses the question “how can the agent efficiently explore the state space during training so that the agent learns the best actions using a smaller number of episodes/samples during training?”
>
> Accordingly, this work typically presents the learning curves during training as the main tool of presentation of the results, for instance, as in  Pathak et al (2017), Bellemare et al (2016).
>
> In contrast, our paper focuses on the following question: “how can we reduce the number/accuracy of the samples the agent takes during the test phase”? (Here, the test phase corresponds to the agent’s behaviour after the training is completed.)
>
> Note the distinction between the behaviour during training (intrinsic motivation) and the test phase (current work).
>
> In particular, if one tries to use the approach of, for instance, Pathak et al (2017)/Bellemare et al (2016)  in the setting of the current paper, one would need to show the behaviour of the agent in the test phase. In particular, one would need to show at each step i) whether the agent chooses to take the observation or skip it  and ii) with which accuracy the agent chooses to take the samples.   On the other hand, in these works, (i)  the agent takes a sample at each step of the environment and  (i) each sample is taken with full-accuracy. Due to (i), one cannot compare with our Scenario B. Due to (ii), one cannot compare with our Scenario A.
>
> (Although the behaviour under noise is illustrated in Pathak et al (2017), the level of the noise cannot be chosen by the agent hence these experiments do not directly provide any observation sampling strategy that is controlled by the agent. )
>
> On description of methodology:
>
> Based on your comments, we have now understood that some clarification is needed on the relationship between intrinsic motivation settings and our work. We have now revised our sentences in multiple places in the article to make this distinction more clear.
>
> On deep RL problems:
>
> We believe that the reviewer refers to the problems where a possibly large multi-dimensional data such as images in games are used as input to the RL algorithm. Assuming this assumption is correct, we agree that it would have been beneficial to have such additional results on the paper. On the other hand, we also think that the current experiments provide clarity in terms of the trade-offs and provide  interesting insights into the standard RL environments.  For instance, our results show that  some of the environments in our experiments are essentially oversampled.  For instance, for MountainCar by Table 1 and Figure 1-c, we observe that it is possible to obtain successful performance even with a sampling rate of 1 / 2, i.e. taking one sample out of two available samples. Given the central importance of these environments as simple ‘sanity checks’ for RL algorithms, we believe that this is an interesting insight.
>
> Comparisons with the methods in literature
>
> As discussed above, we believe that the methods suggested by the reviewer do not provide direct comparisons for the current work.

---

### Official Review · AnonReviewer2 · 2020-10-27
**Interesting setup, but limited experiments**

**Rating:** 6
**Confidence:** 4

**Review:**


In contrast to standard reinforcement learning (RL), the paper investigates the variant where the observation made by the agent about its state has a cost. The authors propose to model the problem as a POMDP with an augmented action space (normal action + observation accuracy) and a new reward function that is defined as the original one penalized by the observation cost. They solve the problem with TRPO in three control domains: mountain car, pendulum, and cart pole.

PROS

I find the research questions asked in the paper interesting. The proposed variation seems to be novel as far as I know. Besides, two scenarios are studied in the paper, which correspond to two extreme cases: continuous vs discrete accuracy.


CONS

The conclusions of the experiments seems to depend on the specific values set notably in Equations (9a-b) and (10). I think a discussion is warranted about how they were chosen. Notably, can the same conclusions be drawn if those values are changed?

I didn't find the information about the policy used in TRPO. Notably, how does it deal with the partial observability?

The paper should be proof-read.

---

> ### Author Response · Authors · 2020-11-15
> **Response to Reviewer 2**
>
> Thank you for your feedback.
>
> On Equations (9a-b) and (10):
>
> The  scaling factor $Q$'s for the noise levels and $\kappa$ values for the reward function are determined empirically by first fixing $Q$ (as a percentage of the full range of the associated observation) and searching for $\kappa$ values that provide satisfactory performance in the original task. Note that the rest of the values are determined by the specifications of the environments in OpenAI Gym. The results depend on the values of $Q$ and $\kappa$. For instance, using larger $\kappa$ puts a larger weight on the reward due to noise. Hence, the agent  prioritizes the reward due to noise instead of the reward from the original environment and, for large enough $\kappa$ values, the agent cannot learn to perform the original task. We have now added discussions on these points in Section 4 and Section A.2.
>
>
> On the usage of TRPO:
>
> Observations are directly fed to the agent without preprocessing. At first sight, it may be surprising that the agent can learn to perform these tasks satisfactorily even if we have not injected any memory to our algorithm, for instance when we only use the current noisy observations for Scenario A. On the other hand, note that in these environments the observations are either noisy versions of hidden states which govern the dynamics or they are closely related to them. From the point of the agent that treats the noisy observations as state this can be interpreted as a configurable MDP (Metelli et al., 2018; Silva et al., 2019) where the agent controls the noise of the dynamics. Hence, the task of the agent can be interpreted as adjusting the noise level in the dynamics which does not necessarily require usage of memory in the decision maker. A related discussion is now added to the article in Section 4.2.  To clarify the setting, we have also revised our explanation of the RL algorithm in Section 4.1.
>
> The article is proof-read and we will continue to do so while updating it during the discussion period.

---

### Official Review · AnonReviewer3 · 2020-10-28
**The paper approaches an interesting problem and proposes a simple, yet reasonable, approach. Unfortunately, the evaluation fails to provide a clear perspective on the potential impact of the proposed approach.**

**Rating:** 5
**Confidence:** 3

**Review:**

= Overview =

The paper proposes a reinforcement learning algorithm that enables an agent to "fine tune" the quality/accuracy of its sensors to its current task. The paper considers a partially observable MDP setting where the agent, besides the control actions, is endowed with a set of "tuning actions" that control the noise in the perception of the different components of the state. Additional reward terms are introduced that discourage the use of "tuning". By enabling the agent to fine tune its perception to the current task, the paper seeks to also investigate the relative importance of different state features in terms of the task.

= Positive points =

The paper is well written and the ideas clearly presented. The ideas seem vaguely related with recent work on "tuning" MDPs [a] and some older work on learning state representations in multiagent settings [b,c], where the agents are allowed to "pay" to have better models or perceptions. The paper proposes the use of similar ideas in a completely different context - to identify relevant information state information in POMDP settings.

= Negative points =

My main criticism is concerned with the particular domains considered, which I believe are too structured to provide a clear understanding of the potential impact of the proposed approach.

= Comments =

I believe that the problem considered in the paper is interesting and follows some recent work on "tuning" MDPs (see ref[a] below). The approach explored is quite simple but that is not an inconvenient per se. My main criticism lies in the fact that -- in my understanding -- the domains selected are too structured to provide really interesting insights.

In particular, all domains considered are classical control problems with essentially deterministic dynamics and full observability. The approach in the paper injects artificial additive noise in the state as perceived by the agent (the paper only provides explicit information regarding the noise in the Mountain Car domain, but I'm assuming that is similar in the other domains).

Now I may be missing something, but it seems to me that, from the agent's perspective, this is equivalent to adding noise to the dynamics of the environment, since the agent treats the observations as state. Therefore, from the agent's perspective, the practical effect of the "sensor tuning" is to actually attenuate the noise in the dynamics, which partly explains the results provided. This renders this work particularly close to those on MDP tuning referred above, and more discussion in this direction would be appreciated.

I think that the paper would greatly benefit from considering richer domains, either where partial observability is a central issue -- such as those from the POMDP literature -- or with richer perceptual inputs --- such as those from game domains.

= References =

[a] A. Metelli, M. Mutti, M. Restelli. "Configurable Markov Decision Processes." Proc. 35th Int. Conf. Machine Learning, pp. 3491-3500, 2018.

[b] F. Melo, M. Veloso. "Learning of coordination: Exploiting sparse interactions in multiagent systems." Proc. 8th Int. Conf. Autonomous Agents and Multiagent Systems, pp. 773-780, 2009.

[c] Y. De Hauwere, P. Vrancx, A. Nowé. "Learning multi-agent state space representations." Proc. 9th Int. Conf. Autonomous Agents and Multiagent Systems, pp. 715-722, 2010.

---

> ### Author Response · Authors · 2020-11-15
> **Response to Reviewer 3**
>
> Thank you for your feedback! Based on your comments, we have now included additional experiments and provided a detailed discussion of  the position of our paper with respect to the line of work you have suggested.
>
> On the interpretation of the observation noise as noise in the dynamics:
>
> This is an interesting interpretation; thank you for bringing it to light!  We agree with your reasoning and that it is possible to interpret the observation noise as a noise on the dynamics which the agent can tune in the given environments.  We have now discussed this interpretation together with the line of work you have suggested in Section 2 and Section 4.2.
>
> Additional experiments:
>
> To have a better understanding of the effect of partial observability, we have investigated the following modification on MountainCarContinuous-v0: Instead of the horizontal position, the agent uses the vertical position as the observation.  The vertical position $y_t \in [0.1, 1]$ is given by $y_t =0.45 \sin(3 x_t)+0.55$. (This equation comes from the specification of the environment in OpenAI Gym) Note that due to $\sin(\cdot)$ function, for most of the $y_t$ values in the range $[0.1, 1]$, there are two possible horizontal position ($x_t$) values. Hence, this environment constitutes a POMDP even without any observation noise. This setting and the results are presented in Section A.4.
>
> References: Thank you for the references. Ref [b]-[c] are now discussed in Section 2. As we have noted above, configurable MDPs and ref[a] are discussed in Section 2 and Section 4.2.
>
> Clarification on the following reviewer’s comment: ``''the paper only provides explicit information regarding the noise in the Mountain Car domain, but I'm assuming that is similar in the other domains''.
>
> As you have pointed out  we have illustrated the procedure with the MountainCar environment in the main text. In the original submission, the noise model template (i.e. modified observation =original observation +noise) were the same for all environments, where we have always used the original observation defined by OpenAI descriptions of these environments as the starting point.  The model for these environments were provided in Section A.2 together with the relevant parameters for the scalings. Together with the publicly available OpenAI descriptions of these environments, (which specify what is given as observation), these provide a complete description of the noise model. For the additional experiments with MountainCar environment with vertical position, the setting and the results are provided separately in Section A.4.

---

### Official Review · AnonReviewer1 · 2020-10-30
**nice visualization scheme, low scientific impact**

**Rating:** 4
**Confidence:** 3

**Review:**

The paper shows how to incorporate an observation cost into RL control problems to assess the inherent value of information in different domains.

I found the paper fun, and well written/edited.

However, I don't see much of a scientific contribution here. The paper says its aim work is to reveal the information structure in the observation space within a systematic framework. So, it's essentially a kind of "ML for scientific visualization" paper. The ML novelty appears small---standard algorithms and test problems are used. The paper isn't really evaluated from a scientific visualization perspective, so it's not clear that it is over the bar from that perspective.

The light shed on some standard test problems ("decisions aren't that impactful when the pole is almost balanced", etc.) are nice, but not really impactful.

Detailed comments:

Related work: I think it would be appropriate to cite Valentina Bayer's "cost sensitive learning" work. I think there's also a "cost observable MDP" model that is very related. The earlier work isn't able to solve these problems as well as the current paper, but the model is very related.

"towards to" -> "towards"

"maximize average" -> "maximize the average"?

Table 1: Use right justification for easier visual comparison.

I'm confused about the state used in the experiments. It's a POMDP, so was there a recurrent network used? Were multiple steps available in the state representation? How were the RL algorithms able to represent and learn the strategy?

"For Mountain car environment all" -> "For the mountain car environment, all"

"following rise" -> "following arise"

"the agents performance" -> "the agents' performance"?

---

> ### Author Response · Authors · 2020-11-15
> **Response to Reviewer 1**
>
>
> Thank you for your comments!  We believe that there is an important misunderstanding in regard to interpretation of our paper as a visualization paper, which we discuss below.
>
> On the interpretation of this paper as a visualization paper:
>
> Although our figures provide visualizations of relative importance of different observations,  this is not our primary aim but a tool to present the optimum strategies and a nice-to-have by-product of our framework. In particular, we note the following in terms of our contributions:
> - In many practical decision making problems, there is a cost associated with obtaining observations. Our proposed framework
> i) allows us to  attack these problems using a RL approach.
>  ii) determines cost-efficient data-acquisition strategies showing  which observations should be prioritized.
>  iii) quantifies the possible performance degradation one may have due to the costly observations (Table~1).
> - The information structure revealed by these cost-efficient strategies leads to scientifically interesting insights. For instance, our results show that some of the environments in our experiments are essentially oversampled from a data collection point of view.   For instance,  for MountainCar by Table 1 and Figure 1-c, we observe that it is possible to obtain  successful performance even with a sampling rate of 1 / 2, i.e. taking one sample out of two available samples. Given the central importance of these environments as simple ‘sanity checks’ for RL algorithms, we believe that this is an interesting insight.
>
> Based on your comments, we have now understood our original presentation was imbalanced in terms of our motivation which has probably contributed to this misinterpretation of the paper as a visualization paper. Accordingly, we have now revised our presentation in ``Section 3.3.1 Motivation.
>
> On learning of the agent:
>
> We have not used RNNs.  Only one step observation is fed as observation to the agent. At first sight, it may be surprising that the agent can learn to perform these tasks satisfactorily even if we have not injected any memory to our algorithm, for instance when we only use the current noisy observations for Scenario A. On the other hand, note that in these environments the observations are either noisy versions of hidden states which govern the dynamics or they are closely related to them. From the point of the agent that treats the noisy observations as state this can be interpreted as a configurable MDP (Metelli et al., 2018; Silva et al., 2019) where the agent controls the noise of the dynamics. Hence, the task of the agent can be interpreted as adjusting the noise level in the dynamics which does not necessarily require usage of memory in the decision maker.
>
> A related discussion is now added to the article in Section 4.2.  To clarify the setting, we have also revised our introduction of the RL algorithm in Section 4.1.
>
> Thank you for the references. We agree that these works fit well to our setting hence we have now included them in our discussions in Section 2 and Section 3.3.1.
>
> Typos are corrected.

---

### Comment · ~Yunhan_Huang1 · 2022-07-14
**Some relevant papers**

I think the following papers can give a theoretical underpinning to the idea the authors proposed:
*  Biedenkapp, André, et al. "TempoRL: Learning when to act." International Conference on Machine Learning. PMLR, 2021.
*  Huang, Yunhan, Veeraruna Kavitha, and Quanyan Zhu. "Continuous-time markov decision processes with controlled observations." 2019 57th Annual Allerton Conference on Communication, Control, and Computing (Allerton). IEEE, 2019.
*  Huang, Yunhan, and Quanyan Zhu. "Self-Triggered Markov Decision Processes." 2021 60th IEEE Conference on Decision and Control (CDC). IEEE, 2021.
* Jacq, Alexis, et al. "Lazy-MDPs: Towards Interpretable RL by Learning When to Act." Proceedings of the 21st International Conference on Autonomous Agents and Multiagent Systems. 2022.
* Wu, Wei, and Ari Arapostathis. "Optimal sensor querying: General markovian and lqg models with controlled observations." IEEE Transactions on Automatic Control 53.6 (2008): 1392-1405.

---

### Decision · Program_Chairs · 2021-01-07
**Final Decision**

**Decision:**

Reject

**Comment:**

The setting and the problem addressed by this paper has been considered as important and interesting to tackle with reinforcement learning. Yet, the reviewers expressed several concerns about this paper. Especially, the lack of comparison to state-of-the-art methods and to the standard visualization methods was a shared concern. The empirical validation also appeared as not ambitious enough. Finally, the novelty in the field of machine learning was also questioned since the paper is mainly about applying existing algorithms to a known problem.